# Performance Monitoring of Mast-Mounted Cup Anemometers Multivariate Analysis with ROOT

**DOI:** 10.3390/s22249774

**Published:** 2022-12-13

**Authors:** Salvatore Mangano, Enrique Vega, Alejandro Martínez, Daniel Alfonso-Corcuera, Ángel Sanz-Andrés, Santiago Pindado

**Affiliations:** 1Centro de Investigaciones Energéticas, Medioambientales y Tecnológicas (CIEMAT), Av. Complutense 40, 28040 Madrid, Spain; 2Instituto Universitario de Microgravedad “Ignacio Da Riva” (IDR/UPM), Universidad Politécnica de Madrid, Pza. del Cardenal Cisneros 3, 28040 Madrid, Spain

**Keywords:** cup anemometer, mast mounted, anemometer recalibration, wind speed measurements

## Abstract

This paper analyzes the field performance of two cup anemometers installed in Zaragoza (Spain). Data acquired over almost three years, from January 2015 to December 2017, were analyzed. The effect of the different variables (wind speed, temperature, harmonics, wind speed variations, etc.) on two cup anemometers was studied. Data analysis was performed with ROOT, an open-source scientific software toolkit developed by CERN (*Conseil Européen pour la Recherche Nucléaire*) for the study of particle physics. The effects of temperature, wind speed, and wind dispersion (as a first approximation to atmospheric turbulence) on the first and third harmonics of the anemometers’ rotation speed (i.e., the anemometers’ output signature) were studied together with their evolution throughout the measurement period. The results are consistent with previous studies on the influence of velocity, turbulence, and temperature on the anemometer performance. Although more research is needed to assess the effect of the anemometer wear and tear degradation on the harmonic response of the rotor’s angular speed, the results show the impact of a recalibration on the performance of an anemometer by comparing this performance with that of a second anemometer.

## 1. Introduction

The degradation of the cup anemometer is a well-known problem in the wind energy industry, as this is the industrial sector that demands the largest unit amount from this instrument in the world (the cup anemometer is an extremely robust and accurate instrument to measure the wind speed that was invented in the XIX century by Robinson; see in [1] a comprehensive literature review regarding the research on this instrument along more than 100 years). As a cup anemometer (see Figure 1) loses performance due to the normal wear and tear process or due to sudden incidents such as lightning, the wind speed measured by the instrument diverges from the real wind speed. Therefore, this effect can be translated into wrong wind turbine operation or inaccurate data when studying the energy production of a specific geographic location, causing a negative impact on the revenue. As mentioned above, this is a new problem neither for the wind energy sector nor in meteorology. Furthermore, around 30% of mast-mounted anemometers returned for recalibration are far from normal operating conditions [2].

At present, the only solution to keep anemometers in proper working condition is to check them by performing frequent recalibrations [3]. However, the process of taking the anemometer to the calibration facility can be unaffordable in terms of costs and time delays. Calibration procedures in the field have been studied as a cost-effective solution to reduce the maintenance of anemometers and the number of recalibrations [4,5]. In addition, to illustrate the interest of the industry in this matter, it should be noted that several patents and inventions have been developed trying to solve this problem [6,7,8,9,10,11,12,13,14].

Leaving aside the maintenance problem (re-calibration, change of parts, etc.), it can be even more important to know, as precisely as possible, when a cup anemometer that is working in the field could require some maintenance.

In the present work, an anomaly-detection process developed at the IDR/UPM Institute [15] is applied to two cup anemometers working in the field (i.e., mounted on a meteorological mast) that will suffer some level of degradation after a quite long period of service. These results are part of a comprehensive research program at the IDR/UPM Institute on cup anemometer performances, which covers a large series of calibration analysis [16], rotor aerodynamics [17,18,19], the effect of climatic conditions [20], or the ageing problem [21].

In the last 2–3 years, the researchers of the IDR/UPM Institute have focused on the experimental performance of the cup anemometer. In this sense, three major initiatives have been generated:Analyze the errors of the output signal generation system of these instruments;Study the possibility of reproducing the performances of the cup anemometer; andAnalyze in the maximum possible detail all the available data coming from anemometers installed in meteorological masts.

These three different studies are not isolated research projects but are clearly related. The study of the opto-electronic signal generation systems of the most commonly used anemometers in the wind energy and meteorology sectors (Thies First Class AdvancedThies Clima, Göttingen, Germany- or those belonging to the A100 series from Vector Instruments-Windspeed Limited, North Wales, UK-) has given rise to two different initiatives to uncover errors in the operation of the cup anemometer and to improve maintenance programs:The first one is the rotational harmonic study (see Section 2 of this paper);The second one has given rise to a methodology that combines two ways of generating the output frequency of the anemometer signal (pulse counting and FFT of this output signal) in order to isolate errors in the opto-electronic system caused by dirt accumulation [22,23].

In addition, the analysis carried out on specific errors, such as the misalignment of parts of this opto-electronic system with the instrument’s rotor rotation axis, should be mentioned [24]. These studies have been fundamental in understanding how these errors can affect the wind speed measurement performed by the cup anemometer. Hence, the importance of developing a simulator that allows to know how the manufacturing processes of these instruments and, more specifically, their accuracy can affect the wind speed measurements [25].

At the same time and given that the available databases on cup anemometer performances are relevant in terms of size, post-processing of these data has been started in order to have a better knowledge of the relationship between the most important variables that affect the performance of a cup anemometer (wind speed, air density, temperature, turbulence, etc.). This paper includes the first results of this post-processing. Finally, it should be mentioned that the present research is part of a quite ambitious project that aims to correlate the results regarding cup anemometer performance obtained by analytical modeling and by testing in a wind tunnel during the last 10 years, with results from the performance of this instrument working in the field.

The present paper is organized as follows: In Section 2 the calibration and post-processing processes are described, together with the relevant theoretical framework. The results are discussed in Section 3, and the conclusions are summarized in Section 4.

## 2. Methodology and Procedure

Thies First Class Advanced cup anemometers have three cups of conical cross-section, 240 mm (9.45″) rotor diameter, and 290 mm (11.42″) overall height, with square wave electrical signal output.

As said in the previous section, the data of two Thies First Class Advanced cup anemometers (hereinafter Anemometer 1, A1; and Anemometer 2, A2) were collected over a period of about three years (from January 2015 to December 2017). Both anemometers were installed on one of the meteorological masts of the Kintech Engineering company. This meteorological mast is placed in a wide-open space in Zaragoza (Spain), without any barriers nearby. Anemometers A1 and A2 are located at 73 m height and 58 m height, respectively. Anemometer 2 was retired from the meteorological mast on 1 July 2016 for recalibration and reinstalled after recalibration on 11 October 2016.

The calibration consisted of determining the slope, A, and the offset, B, of the sensor transfer function (also known as the calibration curve), which is the linear correlation between the wind speed, *V*, and the output frequency of the cup anemometer, *f*:(1)V=Af+B

The calibration was performed at LAC (ISO-17025 accredited laboratory), an IDR/UPM Institute facility. The calibration coefficients of both anemometers are shown in Table 1. The calibration process followed MEASNET recommendations [26,27] (over 13 points and from 4 to 16 m/s wind speed). The facility is an open-circuit, closed-test-section wind tunnel, with the flow in the testing area being extremely uniform (the maximum difference in the wind speed is less than 0.2%).

Data recording started on 13 January 2015 (anemometer A2) and on 4 April 2015 (Anemometer A1) and ended (for both Anemometers A1 and A2) on 29 December 2017. Anemometer output signals were sampled at 5 kHz during 20 s each 10 min (that is, one data set of 20 s per anemometer was taken each 10 min). These measurement rates were imposed by the characteristics of the hardware used and the software programmed.

Sensor outputs were recorded by using a National Instruments^®^ CompactRIO (National Instruments Corp., Austin, TX, USA) system with a 16-bit voltage acquisition module. The CompactRIO systems combine the measurement capability of programmable laboratory equipment, a rugged design, and large data storage capacity, which makes them suitable for long-term field testing.

Some periods of these nearly three years are missing due to anemometer maintenance, anemometer calibration, or failures of the data acquisition system. The other instruments of the mast provide information about date, temperature, output frequency, etc.

For data check, both anemometers were also connected to the Kintech meteorological mast measurement and recording system. In Figure 2, a comparison of the wind speed measurements taken with Anemometer 2 connected to the mast measurement system (i.e., to the data logger that recorded the wind speed each minute) and taken with the CompactRIO system designed for the present study is shown. In the figure, the equivalence of both measurement systems can be observed. The percentage variation of the CompactRIO system measurements, Δ*V*, in relation to the ones from the datalogger are also plotted in Figure 2.

Although the differences seem not to be relevant for most of the points, some of them show quite a large variation. This is mainly produced because the datalogger takes 1 min average wind speed measurements, whereas the CompactRIO system takes a 20 s average within a 10 min period. Additionally, it should also be highlighted that even taking the closest instant points from each measurement system to be compared, it may have some difference between the instants when the measurements were taken by both systems. Each 20 s dataset taken every 10 min period, composed of *n* turns of the cup anemometer rotor, was post-processed to obtain the average output frequency, *f*. After that, each turn was analyzed to extract the average rotation rate, *w*_0,*j*_, and the harmonic terms in that turn, *j*, bearing in mind that in a constant wind speed, the rotation rate of a cup anemometer rotor [14], i.e.,
(2)$$\begin{array}{l} \omega \left(t\right)=\omega_{0}+\omega_{1}\sin\left(\omega_{0}t+\varphi_{1}\right)+\omega_{2}\sin\left(2\omega_{0}t+\varphi_{2}\right)+\omega_{3}\sin\left(3\omega_{0}t+\varphi_{3}\right)\ldots \\ =\omega_{0}+\sum_{i=1}^{\infty }\omega_{i}\sin\left(i\omega_{0}t+\varphi_{i}\right) \end{array}$$
is composed of a constant rotation rate, *w*_0_, and certain harmonic terms.

From the *n* turns in each dataset, the average rotation speed, *w*_0_, was obtained together with its standard deviation; used to obtain a measure of the velocity deviation, *σ_u_*; and defined by standard deviation of the measured wind speed in each turn and the averaged harmonic terms of the rotation, *H_i_*:(3)$$H_{i}={\bar{\omega }}_{i}=\frac{1}{n}{\sum_{j=1}^{n}\frac{\omega_{i}}{\omega_{0}}|}_{j}$$

### 2.1. The Harmonic Analysis Applied to Cup Anemometer Performance

The aforementioned coefficients are related to the harmonic approach to the cup anemometer performance, which has been applied to this instrument, as the rotor anemometer has a periodic behavior working at constant wind speed. That is, both the aerodynamic forces and the performance are periodical, with frequency (and its harmonic terms) equal to the rotation frequency. Some notes are included in the following subsections to describe the theoretical context in which the research of the present paper is framed.

#### 2.1.1. Harmonic Analysis Applied to the Aerodynamic Forces on the Rotor Cups

According to the available literature, the first time that harmonic analysis was used in the study of cup anemometers was in 2013 [18]. In this article, the analysis of cup anemometer performances is approached from the aerodynamics of the cups but establishing a different approach to the one commonly used until then, which was used by researchers such as Schrenk [28], Wyngaard [29], or Ramachandran [30,31].

This approach simplified the aerodynamics of a rotating bowl by assuming (1) that the aerodynamic forces at one position of the cup are similar to those of that same cup, static (i.e., not rotating), and located at a yaw angle equal to that of the rotating cup and (2) that the aerodynamic force at the center of the cup, represented by the normal aerodynamic force coefficient, *c_N_*, took only two values, i.e., *c_d_*_1_ and *c_d_*_2_. See in Figure 3 the coefficient, *c_N_*, measured in static (i.e., non-rotating) cups [32] and the 2-coefficient approach to these experimental measurements.

The Equation that describes the cup anemometer performance is the following:(4)$$I\frac{d\omega }{dt}=Q_{A}+Q_{F}$$
where *I* is the moment of inertia of the rotor, *ω* is its rotation speed, *Q_A_* is the aerodynamic torque on the rotor, and *Q_F_* is the friction torque acting on the anemometer’s saft. In the above mathematical expression, it is reasonable to assume that the friction torque is irrelevant compared to the aerodynamic torque. Therefore, if this Equation is averaged along one turn of the rotor, the averaged aerodynamic force on each cup is equal to zero, and the following Equation is derived [33]:(5)$$Q_{A}=0=\frac{1}{2}\rho S_{c}R_{rc}\left({\left(U_{\infty }-\omega R_{rc}\right)}^{2}c_{d1}-{\left(U_{\infty }+\omega R_{rc}\right)}^{2}c_{d2}\right)$$
where *ρ* is the air density, *S_c_* is the front area of the cups, *R_rc_* is the cups’ center rotation radius, and *U*_∞_ is the wind speed. From this Equation, a quite accurate solution can be obtained:(6)$$K=\frac{U_{\infty }}{\omega R_{rc}}=\frac{k_{d}+1}{k_{d}-1};\;k_{d}=\sqrt{\frac{c_{d1}}{c_{d2}}}$$
where *K* is the anemometer factor.

If a Fourier approach instead of the 2-coefficient approach is selected (see Figure 3), the averaged aerodynamic torque on each cup, equal to zero, can be expressed as [15]
(7)$$Q_{A}=0=\frac{1}{2}\rho S_{c}R_{rc}V_{r}^{2}\left(\theta \right)c_{N}\left(\alpha \left(\theta \right)\right)$$
where *V_r_* is the relative-to-the-cup wind speed:(8)$$V_{r}^{2}\left(\theta \right)=U_{\infty }^{2}+{\left(\omega R_{rc}\right)}^{2}-2U_{\infty }\omega R_{rc}\cos\left(\theta \right)$$
and *θ* is the angular position of the cup in relation to the anemometer (and not in relation to the wind speed direction), *α* (see Figure 4).

The relationship between these two angles, *α* and *θ*, depends on the ratio between the wind speed and the rotation speed and is defined by [15,19]
(9)$$\tan\left(\alpha \right)=\frac{K\sin\left(\theta \right)}{K\cos\left(\theta \right)-1}$$

Additionally, the normal aerodynamic force coefficient, *c_N_*, can be approached by a Fourier series:(10)$$c_{N}\left(\alpha \right)=c_{0}+c_{1}\cos\left(\alpha \right)+c_{2}\cos\left(2\alpha \right)+c_{3}\cos\left(3\alpha \right)+\ldots$$

See in Figure 3 the 1-harmonic Fourier series approximation:(11)$$c_{N}\left(\alpha \right)=c_{0}+c_{1}\cos\left(\alpha \right)$$
to the experimentally measured aerodynamic force coefficient, *c_N_*, of a cup. Bearing in mind the relation between *α* and *θ*, the following approach was proposed [15,19]:(12)$$\cos\left(\alpha \right)=\eta_{0}+\eta_{1}\cos\left(\theta \right)+\eta_{2}\cos{\left(\theta \right)}^{2}+\eta_{3}\cos{\left(\theta \right)}^{3}$$
in which the coefficients of the right side are defined as
(13)$$\begin{array}{l} \eta_{0}=\frac{-1}{\sqrt{1+K^{2}}};\;\eta_{1}=\frac{K}{\sqrt{1+K^{2}}}-\frac{1}{K^{2}-1};\\ \eta_{2}=\frac{1}{\sqrt{1+K^{2}}};\;\eta_{3}=\frac{K^{2}}{K^{2}-1}-\frac{K}{\sqrt{1+K^{2}}} \end{array}$$

Finally, the following Equation can be derived as an alternative to Equation (6) [15,19]:(14)$$0=\left(1+\frac{1}{K^{2}}\right)\left(1-\frac{1}{2}\frac{c_{1}}{c_{0}}\frac{1}{\sqrt{1+K^{2}}}\right)-\frac{1}{4}\frac{c_{1}}{c_{0}}\frac{1}{K}\left(\frac{K}{\sqrt{1+K^{2}}}+\frac{3K^{2}-4}{K^{2}-1}\right)$$

The above Equation is quite accurate. Nevertheless, some improvements can be made if, in the Fourier approach, two important facts are considered [34]:Due to the cup rotation, the normal aerodynamic force coefficient, *c_N_*, is not symmetrical with regard to *θ* = 180°; andThis normal aerodynamic force (vector *N* in Figure 4) is not applied at the center of the cup, being displaced depending on the angular position of the cup [1].

At present, only a theoretical approach has been carried out, and the results are quite relevant in terms of accuracy regarding experimental data [34].

#### 2.1.2. Harmonic Analysis Applied to the Cup Anemometer Performance (i.e., the Rotational Speed)

As abovementioned, the rotation speed of a cup anemometer working in constant wind speed, *ω*, can be decomposed into an average term, *ω*_0_, and some additional minor Fourier harmonic terms (see Equation (2)). This periodic distribution of the rotation speed is caused by the shape of the anemometer’s rotor, which is composed of three cups, each one responsible for a paired acceleration and deceleration in each turn of the rotor.

Leaving aside the average value, *ω*_0_, which is related to the wind speed with Equation (1), the harmonic terms more relevant to the wind speed are the first one, *ω*_1_ (related to friction and other imperfections that may produce one perturbation per turn of the rotor) and the third one and its multiples, *ω*_3_, *ω*_6_… (which are related to the aerodynamics of the cups that causes three perturbations per turn of the rotor). Finally, the other harmonic terms are produced by other phenomena such as turbulence or the interaction of the rotor with wakes in the flow due to upstream objects.

The harmonic distribution can be extracted from the output signal of the cup anemometer. The more advanced cup anemometers normally give a squared-pulse signal of *N* pulses per turn [16]. From these pulses, it is possible to extract the rotation speed along one turn of the rotor and also the harmonic terms. See in Figure 5 the output signal, *u_out_*, of a Climatronics 100075 cup anemometer; the rotation speed normalized, *ω*/*ω*_0_; and the harmonic terms extracted from this tree-wave distribution, *ω _i_*/*ω*_0_.

Variation of harmonic terms has been used to identify different situations in which the rotor was damaged [35]. This is particularly relevant, as a cup anemometer with the rotor damaged may rotate even faster than a non-damaged one, or a cup anemometer without one cup that is working in flag position (that is, locked into a static equilibrium position) may still produce a train of pulses proportional to the wind position (and, as a consequence, make difficult to identify this wrong state of the sensor) (see Figure 6). It should also be said that a new method for identifying errors by comparison of two different ways of extracting the output frequency of cup anemometers has been proposed recently [23].

### 2.2. Post-Processing the Data with ROOT

Data reduction and statistical analysis were implemented with the ROOT (https://root.cern/, accessed on 6 November 2022) [37,38,39] software framework, which uses the anemometers’ data that are saved in ROOT format files. ROOT is a free and open-source scientific software toolkit developed at CERN (https://home.cern/, accessed on 6 November 2022) (*Conseil Européen pour la Recherche Nucléaire*), which is an European research center that operates a particle physics laboratory located near Geneva, Switzerland. This software framework is widely used by experimental collaborations, such as particle physics, astrophysics, and high-energy astronomy experiments, to study the data and their interpretation [40,41,42,43].

ROOT is an object-oriented data analysis framework, written in C++ and originally developed by René Brun and Fons Rademakers in 1994. It contains modular software designed for statistical data processing and analysis and includes sophisticated visualization interfaces, including high data storage performance. The data structures for ROOT format files for large data sets with huge files can be compressed easily and have efficient methods to traverse the hierarchical data format files, saving substantial amounts of time during the data analysis. However, even for small datasets, ROOT is an appropriate solution, as histograms can be used to represent a dataset and to obtain their statistical information.

## 3. Results

Figure 7 and Figure 8 show the values of the first and third harmonic terms, *H*_1_ and *H*_3_ (see Equation (3)), of both anemometers as a function of temperature. In the graphs corresponding to these figures, the values are plotted for mean rotational speeds greater and lower than 20 rad/s. First, it can be observed that the first harmonic term is higher for low speeds; i.e., the effect of the disturbances that occur once per turn of the rotor (in the present case, the friction produced on the shaft of the anemometer in which the rotor is fitted) increases for low rotation speeds. This produces a drop in speed, which in turn increases the third harmonic since the rotor accelerations produced by the aerodynamic forces on each cup are more relevant in relation to the average rotational speed. This increase in the third harmonic term at low wind speed has already been reported [14].

Additionally, the graphs in Figure 7 indicate that the effect of the temperature increases the first harmonic term (quite slightly in the case of Anemometer 1). This is produced by a decrease in air density, which reduces proportionally the aerodynamic torque on the anemometer rotor. Considering Equation (4), which describes the performance of the cup anemometer, it is possible to assume that a decrease in the aerodynamic torque is reflected in an increased relevance of the friction torque on the anemometer’s performance. Furthermore, bearing in mind the relationship between the friction forces and the first harmonic term indicated previously, we can assume that the first harmonic term will increase with increasing temperature.

In addition, the decrease in the air density produced by the increase of the temperature reduces the aerodynamic forces on the cups (which create the aerodynamic torque), as they depend on the dynamic pressure. The accelerations/decelerations produced by the three cups in one turn are then reduced, and therefore, the third harmonic term is reduced, as can be observed in Figure 8.

The effect of the non-dimensional wind speed deviation, *σ_u_*/*u*, where *u* is the 20 s wind speed average, on the first and third harmonic terms of both anemometers studied is shown in Figure 9 and Figure 10. Up to a certain limit, this parameter can be considered a proxy for atmospheric turbulence. The results seem to indicate that the turbulence increases the value of the first harmonic term, with this effect being more noticeable for lower values of the wind speed. Additionally, the results show a lower effect of the turbulence on the first harmonic term related to Anemometer 1 when compared to the first harmonic term related to the Anemometer 2. The effect of the turbulence on the third harmonic term tends to decrease asymptotically this value.

Although no effect on the harmonic terms regarding both anemometers has been observed in the results, the calibration of Anemometer 2 was reflected in the wind speed measurements if it is compared with the measurements taken by Anemometer 1. In Figure 11, the difference between the wind speed measured by both anemometers is plotted as a function of the number of the data order (entries) organized from the first to the last. The event that shows the recalibration of Anemometer 2 is plotted with a vertical dashed line. It can be observed that after the calibration, both anemometers seem to measure much closer in wind speed. A similar result when comparing the performance of two cup anemometers working at the same time has been previously shown by other researchers [44,45,46].

The effect of air pressure and humidity on the instruments’ performance was not analyzed, as changes in both variables at the location where the anemometers were installed do not sufficiently alter air density so as to require a correction of the measured wind speed.

### Discussion

The data observed at a temperature around 15 °C, namely *ω*_1_/*ω*_0_~6·10^−3^ and *ω*_3_/*ω*_0_~1.2·10^−2^, indicate a behavior of the anemometers similar to that already recorded in commercial anemometers [35,47].

Although this work has not recorded any event that could cause a serious alteration of the harmonics of the rotational speed of the anemometers studied (breakage of one of the cups, asymmetric accumulation of dirt in them, change in the mechanical properties due to lightning strikes, dirt in the opto-electronic systems, etc.), it is true that it has been possible to detect the change produced by the recalibration of one of them. Regarding the variables analyzed, the results are in agreement with those measured and simulated in previous works. However, it is noted that it may be essential to develop more accurate measurement systems, i.e., a higher sampling rate, to measure harmonic phenomena more accurately, as shown in [48].

As abovementioned, the effect of possible changes in atmospheric pressure and humidity has not been studied. Both variables have a slight effect on air density that can reasonably be considered of second order compared to the effect of temperature, which affects not only air density but also friction torque (see Equation (4)). Finally, it is also fair to say that in the case of extreme atmospheric phenomena such as tornadoes or hurricanes, which can create large variations in air density, it is possible to correct the velocity by multiplying it by the root of the quotient between the air density during the calibration of the instrument and the air density during the measurement process. This procedure is based on the work of Bauer and Mason [49], who studied the performance of an instrument very similar to a cup anemometer operating in both water and air, and it has been used by researchers of the IDR/UPM Institute to test the possible use of cup anemometers to measure wind speed in stratospheric balloon missions [50,51].

## 4. Conclusions

In the present paper, the first results from the data post-processing of the measurements taken by two Thies First Class Advanced cup anemometers across almost three years are described. Post-processing was carried out with the ROOT framework. The most relevant results of the present work are the following:The first harmonic term of the rotation speed of a cup anemometer, *H*_1_, decreases with both temperature, *T*, and rotation speed, *ω*_0_ (i.e., wind speed);The third harmonic term, *H*_3_, tends to decrease with the air temperature, *T*, as the aerodynamic forces on the cups depend on the air density. Additionally, larger values of rotation speed, *ω*_0_ (i.e., wind speed), seem to produce lower values of this harmonic term;The wind speed turbulence seems to increase the first harmonic term, *H*_1_, with this effect being more noticeable at lower values of rotation speed, *ω*_0_ (i.e., wind speed);On the contrary, the third harmonic term, *H*_3_, tends to decrease with the wind speed turbulence;Additionally, the effect of a cup anemometer recalibration was observed by comparing its performance (before and after recalibration) with the performance of the other anemometer located close to it.

The results described in this paper fall under a larger project or research framework that aims to correlate the results regarding the cup anemometer performance obtained during the last decade at the IDR/UPM institute, with results from the performance of this instrument working in the field. This research framework is oriented towards the improvement of the maintenance of these instruments, which is of a paramount importance in terms of economic revenue in the case of properly adjusting wind turbine maximum efficient points or in the case of investment in wind farm allocation.

## Figures and Tables

**Figure 1 sensors-22-09774-f001:**
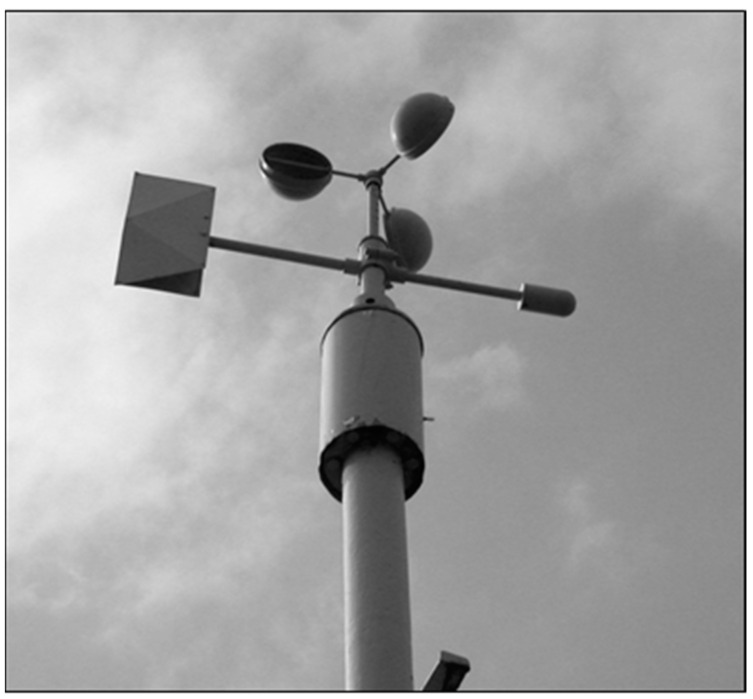
Cup anemometer mounted on a meteorological mast. Photo courtesy of Agencia Estatal de Meteorología (AEMET), Spain. http://hdl.handle.net/20.500.11765/11425 (accessed on 12 December 2022).

**Figure 2 sensors-22-09774-f002:**
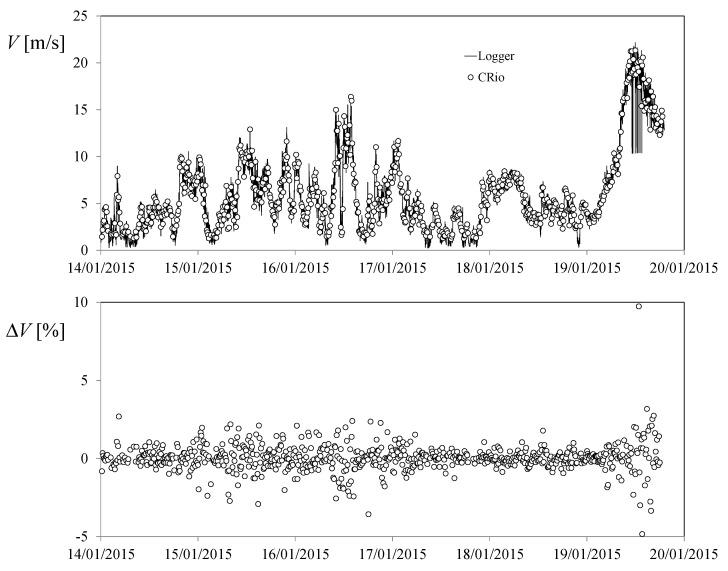
Comparison between the wind speed measured by the CompactRIO system and the Kintech data logger, both connected to the Anemometer 2 output. In the bottom graph, the percentage variation of the CompactRIO measurements in relation to the data logger are plotted.

**Figure 3 sensors-22-09774-f003:**
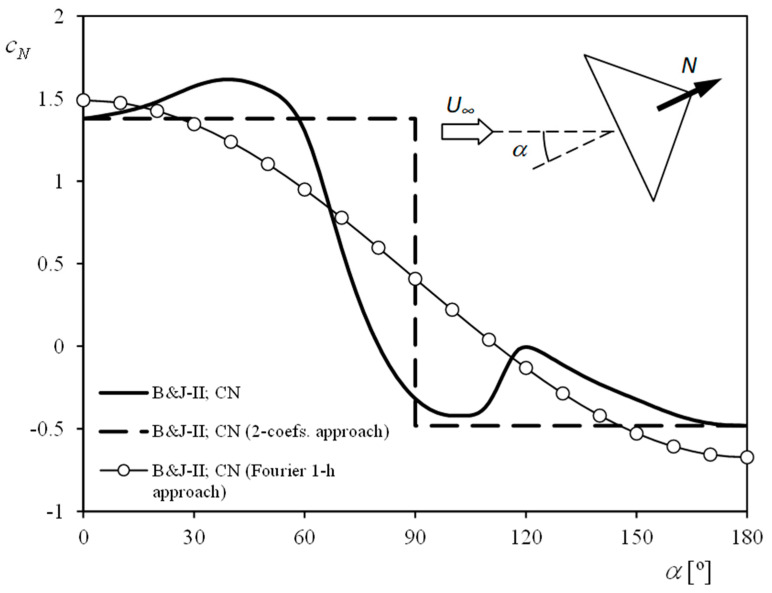
Coefficient *c_N_*, measured on static cups by Brevoort and Joyner [32] in relation to the cup yaw angle (see the sketch added to the figure). The 2-coefficient approach and the 1-harmonic Fourier approach to these experimental measurements are included in the graph.

**Figure 4 sensors-22-09774-f004:**
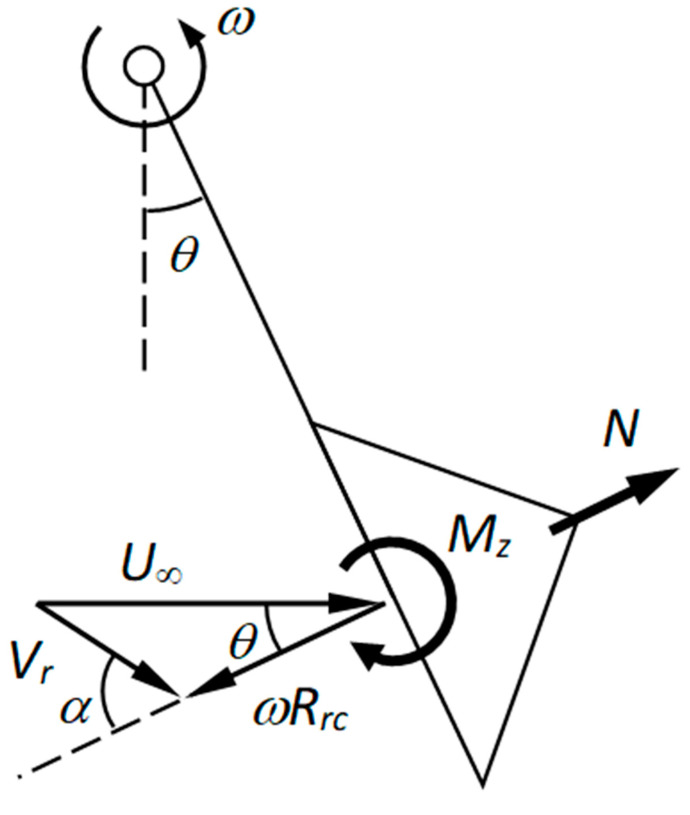
Sketch that shows the wind speed, *U*_∞_, the relative-to-the-cup wind speed, *V_r_*, and the two angles involved: the one, *α*, that indicates the wind speed direction in relation to the rotating cup and the other one, *θ*, which indicates the angular position of the cup (i.e., the angular position of the rotor).

**Figure 5 sensors-22-09774-f005:**
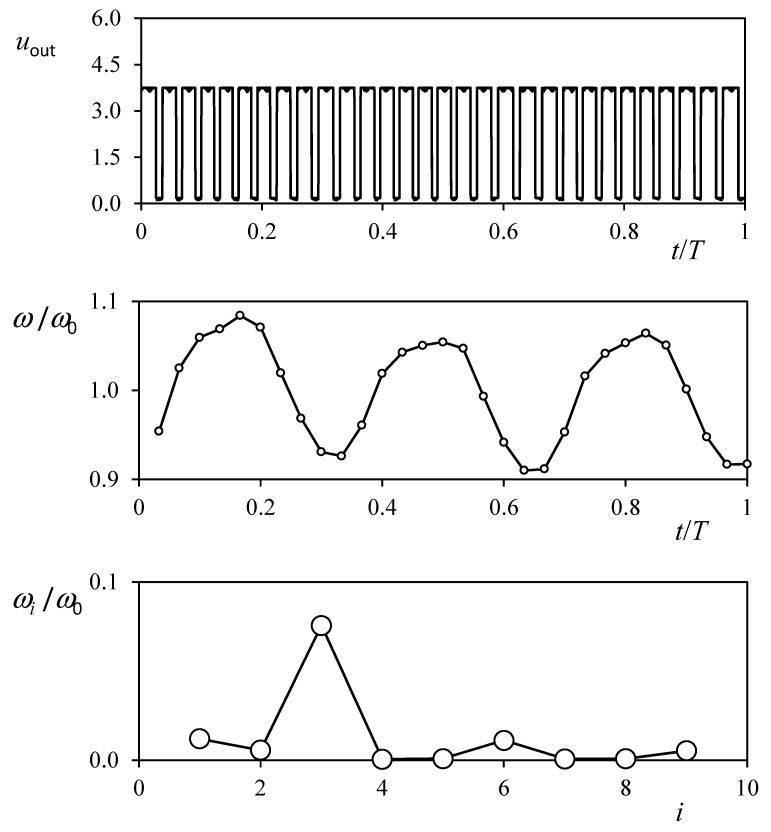
Voltage output signal, *u_out_*, from a Climatronics 100075 cup anemometer (**top**). The rotation rate derived from that signal is included in the (**middle** graph), whereas the Fourier series extracted from the rotation rate is included in the (**bottom** graph) [36].

**Figure 6 sensors-22-09774-f006:**
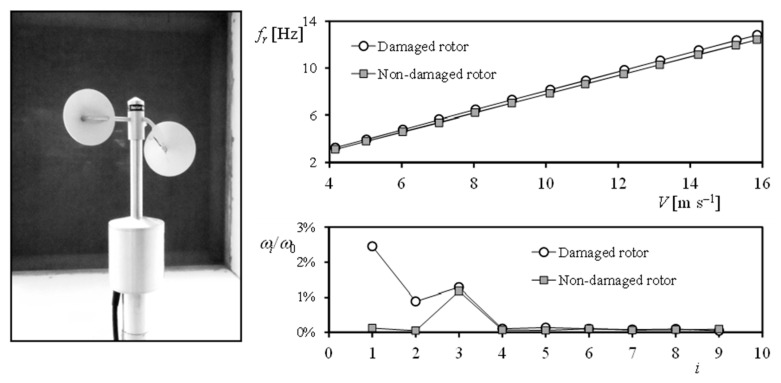
Damaged A100 LK cup anemometer (**left**). Calibration curve of this anemometer compared to the one of the same anemometer body equipped with a non-damaged rotor (**top right**). Fourier series decomposition of the aforementioned cup anemometer rotation rate along one turn of the rotor (**bottom right**) [36].

**Figure 7 sensors-22-09774-f007:**
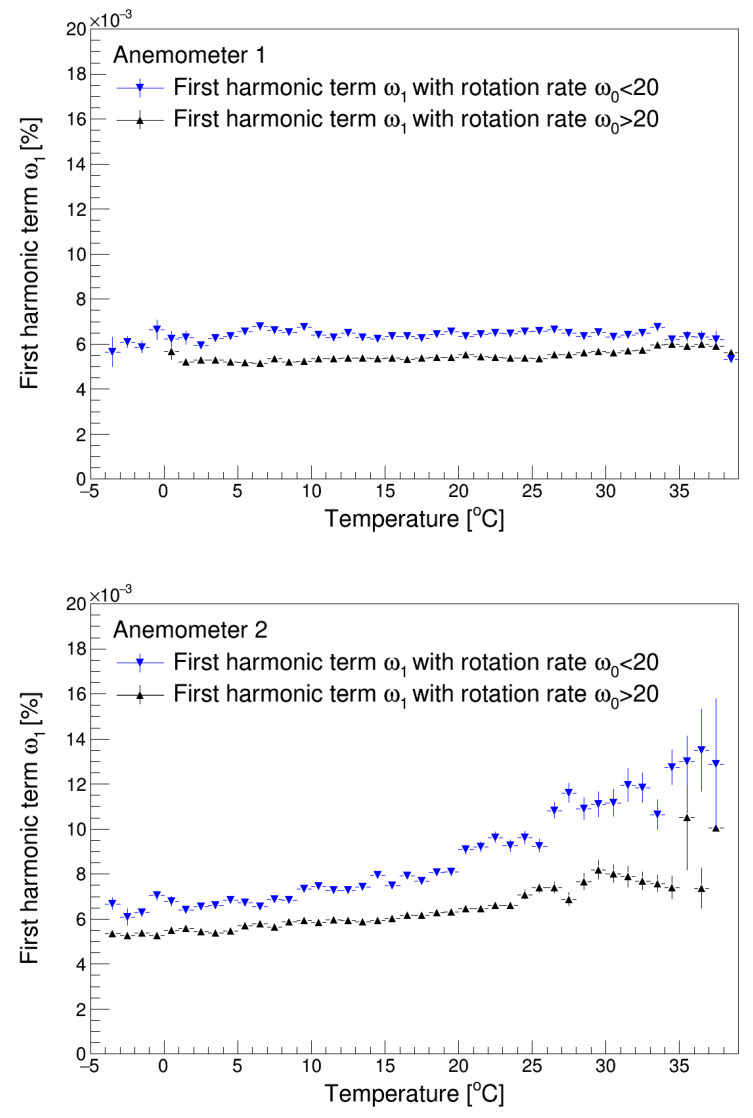
First harmonic terms of Anemometer 1 (**top**) and Anemometer 2 (**bottom**) as a function of temperature.

**Figure 8 sensors-22-09774-f008:**
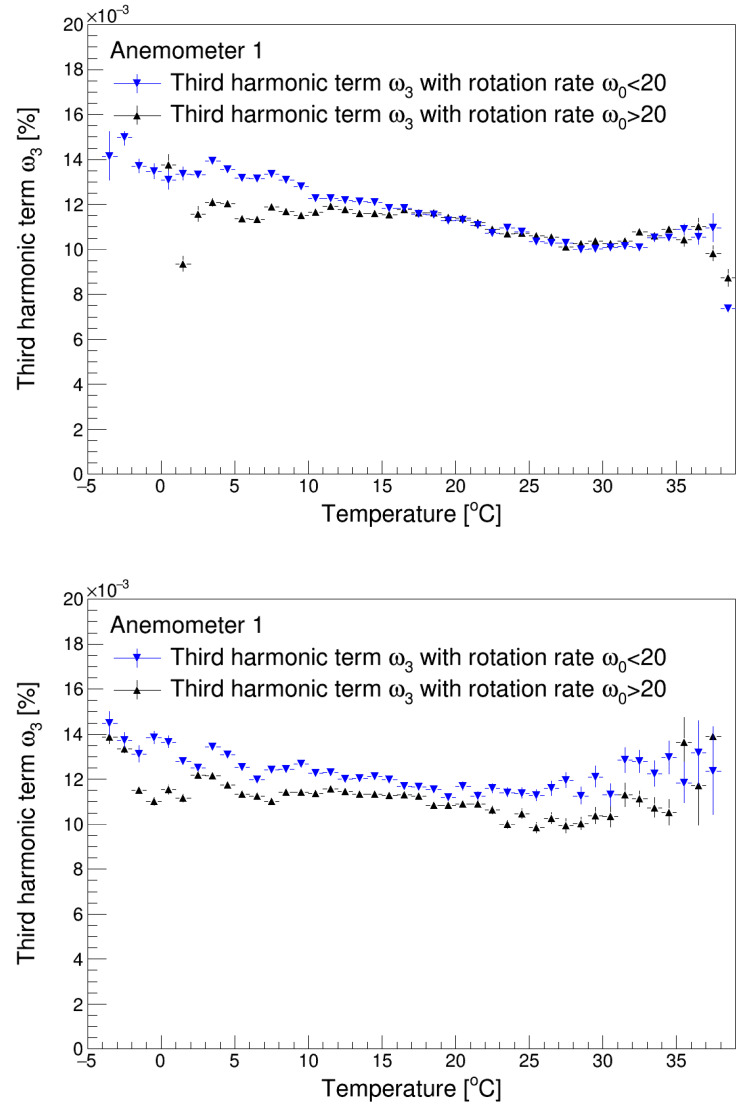
Third harmonic terms of Anemometer 1 (**top**) and Anemometer 2 (**bottom**) as a function of temperature.

**Figure 9 sensors-22-09774-f009:**
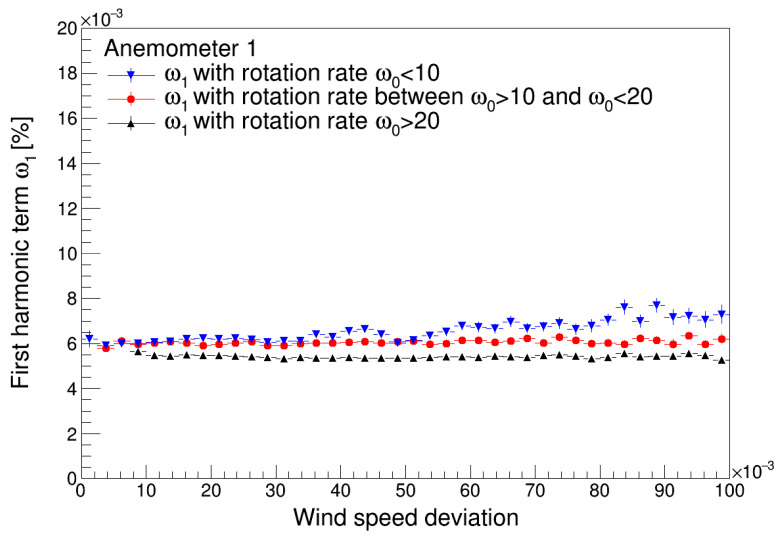
First harmonic terms of Anemometer 1 (**top**) and Anemometer 2 (**bottom**) as a function of the non-dimensional wind speed deviation.

**Figure 10 sensors-22-09774-f010:**
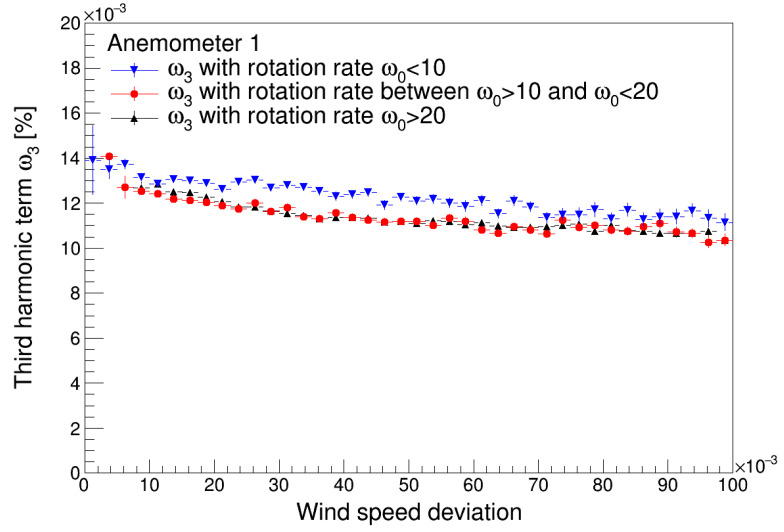
Third harmonic terms of Anemometer 1 (**top**) and Anemometer 2 (**bottom**) as a function of the non-dimensional wind speed deviation.

**Figure 11 sensors-22-09774-f011:**
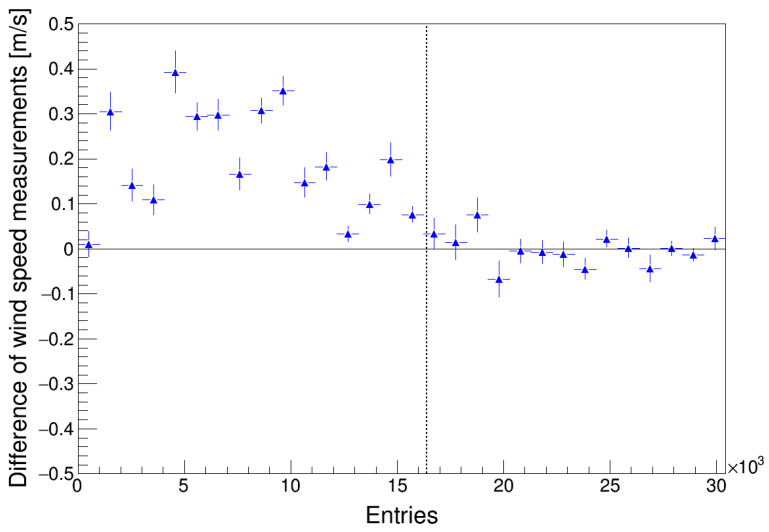
Difference between the wind speed measured by both anemometers is plotted regarding the data (entries) organized from the first to the last. The Anemometer 2 recalibration event (see Table 1) is plotted with a vertical dashed line.

**Table 1 sensors-22-09774-t001:** Calibration coefficients of Anemometer 1 and Anemometer 2 and dates when they were measured.

Sensor	Date	A (m)	B (m/s)
Anemometer 1	19 January 2015	4.6054499·10^−2^	0.26775189
Anemometer 2	14 November 2014	4.6201745·10^−2^	0.24465440
Anemometer 2	15 July 2016	4.6222883·10^−2^	0.26985310

## Data Availability

Not applicable.

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
