# Peer review of "Performance Monitoring of Mast-Mounted Cup Anemometers Multivariate Analysis with ROOT"

_sensors, 2022, doi:10.3390/s22249774_

Round 1

Reviewer 1 Report

- P.1, Section 1, Introduction: This section must improve to 1÷11/2 pp.

- P.2, Section 2: Put more details about anemometers;

- P.2, Eq.(1): Correct it; What other statistical quantities can be used in the analysis?

-P.9, Section 3: Apart from the wind speed, what are the other quantities measured? What sensors were used?

- Section 3: What are those experimental measurements useful for? Improve the explanations. Have you studied the effect of humidity and atmospheric pressure?

- P.14: Put a new section Discussion (1-2 pp.) with modelling and experimental results (Pp. 10-14: Figs.7-11).

- Why is this research important?

- P.14, Section 4, Conclusions: What are the research directions resulting from this article?

Author Response

please, read enclosed file

Reviewer 2 Report

This paper analyzes the field performance of two mast-mounted cup anemometers with ROOT software. It provided useful information. However, some problems need to be solved.

1.In Abstract, what are the meaning of “ROOT” and “CERN”? Please explain at the first time.  

2.The calibration is the key to ensuring the performance of cup anemometer.  Please introduce the method of calibrating the process, not only cited as “ at the 80 IDR/UPM Institute facility can be found in [15].”

3.In Figure 2,” Comparison between the wind speed measured by the CompactRio system and the 100 Kintech data logger, both connected to the Anemometer 2 output.”. It is very difficult to observe the difference between two systems: the CompactRio system(A) and the 100 Kintech data logger (B),. Please use the difference (A-B) as the ordinate.

4. What is the difference between Figure 3 and Figure 3x? I cannot find Figure 3x?

5. In Figure 6x, as i=1,2,3, the characteristics of the damaged rotor are easy to find. However, what happened to the non-damaged rotor at i=3?

6.The quality of Figures 7,8,9,10, and 11, need to be improved.

7. Line 260-262, “Additionally, the graphs from Figure 7 indicate that the effect of the temperature increases the first harmonic term (quite slightly in the case of the Anemometer 1). This is produced by a decrease in air density, that reduces proportionally the aerodynamic torque on the anemometer rotor.”. That is the results in a visual way, not the scientific method. Why does the temperature only affect the first harmonic term as T> 0 for Anemometer 1 and affect the first harmonic term in all temperature ranges for Anemometer 2? Why the effect range of temperature is different between two Anemometers?  

8. In the section “Results”, lines 249-291, all results are observed in visual ways, not the statistical method. Please use the collected data and statistical methods (such as regression or ANOVA) to evaluate the affecting factors. The title of this paper mentioned the word” Statistical analysis”. The content of this paper lack of statistical techniques

Author Response

please, read enclosed file

Round 2

Reviewer 1 Report

- P.3, R.116, Eq.(1): Correct it;

- P.10, Section 3: Improve the explanations. Have you studied the effect of humidity and atmospheric pressure?

- P.14: Improve the Discussion section (to 1 p.).

Author Response

please, find our answer in the attached file

Reviewer 2 Report

The revised paper has been improved significantly.

Author Response

(The authors gave the same response as above.)
